# The Relation between Outdoor Microclimate and People Flow in Historic City Context the Case Study of Bologna within the ROCK Project

Andrea Boeri, Danila Longo [ID], Kristian Fabbri *[ID], Rossella Roversi [ID] and Saveria Boulanger [ID]

Department of Architecture, University of Bologna, 40136 Bologna, Italy
* Correspondence: kristian.fabbri@unibo.it

**Abstract:** Life quality in urban contexts is related to several interconnected factors. Lots of innovative technologies allow for the gathering of real-time information, which is helpful for analysing and interpreting significant urban dynamics and citizens' behaviours. The presence of people in outdoor environments, particularly for social and recreational purposes, can be considered as a qualitative indicator, giving evidence of a living environment. The relationship between urban areas and the climate context has been addressed in recent years by the scientific literature. However, these studies did not report the direct correlation between people's presence and outdoor thermal comfort in outdoor spaces. The aim of this paper is to assess whether the presence of people in outdoor public spaces, detected through human presence sensors, can be associated with outdoor microclimatic conditions (both with on-site measurement and software simulation) and outdoor thermal comfort indicators (as physiological equivalent temperature). The question is whether there exists a direct correlation between outdoor microclimate in public spaces and people's presence, and if a public event plays a role in changing it. The research compares on-site measurements of physics variables (e.g., air temperature) and people's presence with outdoor microclimate maps by Envi-met. The case study, carried out in the framework of the H2020 project ROCK—Regeneration and Optimization of Cultural Heritage in Creative and Knowledge cities, focuses on two squares located in Bologna's historic city center. The conclusions show that public events are the main deciding factor influencing square crowding; nevertheless, the study reveals a relationship between thermal comfort and the number of people.

**Keywords:** outdoor microclimate; pedestrian comfort; people flow; Envi-met modeling; microclimate validation; people presence; outdoor microclimate map

## 1. Introduction

Cities and the built environment are the contexts in which most human activities take place, both inside and outside buildings. In particular, the use of outdoor space depends on multiple factors, e.g., environmental (climate and seasons), architectural (open space configuration such as squares, parks, roads, etc.), urban features and management (urban context, rural, historic center, etc.), economic factors (presence of activities, people flow) and social factors (public events, attractive places, etc.).

The quality of life in urban environments [1] is closely related to the above-mentioned elements and their possible combinations. Among them, some are directly measurable, such as climate and the physical and dimensional characteristics of architectural settlements; others are indirectly measurable, such as crowding, traffic and pollution and, finally, others cannot be measured, i.e., people's affection and care for certain city areas. Such affection is, in most cases, related to cultural, symbolic and identity values (as for historical centers) and is linked to local traditions. In other terms, the presence of people in outdoor environments, particularly for social and recreational purposes, represents a qualitative indicator, providing evidence of a living space.

### 1.1. Outdoor Microclimate Simulation with Envi-Met Software

In recent decades, the temperature increase due to climate change has been perceived with particular intensity in urban areas, where compact building configurations and the low presence of vegetation have exacerbated the urban heat island (UHI) phenomenon [2]. In the Mediterranean context, in particular, the adoption of solutions that allow for improving outdoor thermal comfort conditions in the summer period is needed so that the temperature increase does not compromise the livability of outdoor spaces and the health conditions of the most vulnerable population groups.

The relationship between urban areas and climate has been addressed in recent years by the scientific literature, passing from the territorial and climatic level to the one addressing the city and the urban microclimate [3–7]. The research takes advantage of methodologies and tools that make outdoor microclimate simulations possible, allowing for the investigation of different mitigation approaches to the urban heat island phenomenon and the study of the geometric configuration of urban spaces in relation to the presence of vegetation and the characteristics of materials.

The Envi-met simulation software (www.envi-met.com, accessed 2 May 2023) is one of the most widely used tools for these purposes. It is a three-dimensional nonhydrostatic microclimate model that uses the fundamental laws of fluid dynamics and thermodynamics to calculate and simulate the climate in urban areas. The typical grid resolution of 0.5–10 m in space with a time step of 1–5 s allows for analysing interactions between buildings, soil vegetation and the atmosphere at different scales [8]. The Envi-met software proved to be a valid tool for supporting the design of interventions that involve an improvement in outdoor space comfort conditions [4]. Moreover, thanks to the use of outdoor microclimate maps (OMMs), the results of the simulations can be translated in a clear and easily understandable way, even for nonexpert users, in order to support comfort zones and microclimatic variations at specific sites [9].

The adoption of microclimatic simulations as a tool for decision-making processes can be used in different areas of investigation and intervention: archaeological sites [10], historic centres with tourist interest [11], newly built districts [12] and redevelopment of existing neighbourhoods [13]. Some authors adopt a regression model of neutral adaptive thermal comfort [14,15].

Some studies, carried out with Envi-met, highlight the close correlation between the architectural and geometric configuration of the built environment and urban microclimate conditions [16]. The urban form, intended for buildings' sizes and their in-between spaces, has a strong impact on the average radiant temperature and on the surface shading pattern. At the same time, it influences wind speed, favouring or sometimes blocking the flow of air [17–19]. Mahgoub et al. underlined the need to develop an open platform for suitable urban space in specific climatic conditions, able to promote an improvement in perceived comfort conditions [20]. Other studies analysed recurrent urban forms in order to identify the spatial characteristics that best respond to comfort-condition needs. Urban canyons [21,22] and courtyard blocks [23–25] are investigated in relation to the height/width ratio (H/W), to the sky view factor (SVF) and to their orientation. Martinelli and Matzarakis, for example, studied the H/W ratio at different latitudes in relation to the microclimatic comfort of court spaces located in Italy. As a result, an H/W ratio of 4:5 or 5:5 is considered more suitable for warmer climates, while an H/W ratio of 3:5 or 4:5 can be effective in more temperate climates [26].

Other research supports the study of building shapes and the relationship between full and empty spaces in the implementation of mitigation measures in urban contexts, for example, the use of urban greening as a strategy for the reduction of the heat island phenomenon. As suggested in [27–29], the selection and positioning of trees, green surfaces and infrastructures, thanks to the evapotranspiration effect and the low vegetation capacity to absorb and retain heat, can lead to temperature reductions (cooling effects), with positive effects on the urban microclimate. These effects, however, must be related to airflow impact on vegetation. The increase in trees, if not carefully studied, can lead to a decrease in airflow,

bringing an increase in the concentration of pollutants at the local scale [30]. Among the greening solutions, Berardi investigates the dual benefit deriving from the use of green roofs in building renovations [31] thanks to the combined use of microclimatic simulations (performed with Envi-met) and building energy simulations (performed with Energy-Plus).

The above-mentioned mitigation techniques, when used individually or jointly, lead to a variation in the outdoor microclimate and, consequently, to an improvement in outdoor thermal comfort perception and a reduction in people's health risks [32–34]. As shown in the literature review, the use of microclimatic indicators (air temperature, relative humidity and wind speed) combined with thermal comfort indicators—predicted mean vote (PMV), physiological equivalent temperature (PET) and universal thermal climate index (UTCI)—represents an important resource for public space designs [35–38]. In order to allow thermal comfort investigations in outdoor spaces, Envi-met software is a valid resource. In particular, the combination of the Envi-met model with LadyBug allows for viewing the spatial distribution of the predicted mean vote (PMV) index [39] and, similarly, the combination of Envi-met and HURES [40] or Rayman [41] allows for mapping the universal thermal climate index (UTCI), a useful index for the analysis of urban and landscape planning and design on human thermal comfort. In order to verify Envi-met accuracy and reliability, some studies carried out the model validation through on-site measurement of microclimatic conditions (air temperature, relative humidity and air speed) [16,25,42]. Other research used alternative methodologies: Fabbri and Costanzo calibrated the model through surface temperature measurement with infrared images taken by a drone [43], while Pirini et al. used the combination of Envi-met and TRNSYS software to improve model accuracy when measuring outdoor comfort during the night [44,45].

### 1.2. Measurement of People Flow in Open Space

The increasing use of innovative technologies in urban contexts allows for the collection of a large amount of data and information in real-time that, if properly processed, can be useful in the analysis and understanding of some particularly complex urban dynamics [46–48]. In fact, these data reflect the behaviours and habits of citizens and can be exploited by policymakers, administrations, planners and service designers to redesign public spaces, services and infrastructures using logic that puts at the centre the needs of communities that habitually benefit from them [48,49].

The possibility of evaluating the impact and validity of the interventions and strategies implemented through the use of instruments for measuring people's presence in public spaces was central to the H2020 project ROCK—Regeneration and Optimization of Cultural Heritage in Creative and Knowledge cities (GA No. 730280) [50]. The project, which lasted four years and ended in December 2020, aimed to enhance the regenerative capacity of cultural heritage (CH) in the urban context. ROCK has supported the transformation of some areas located in central positions towards smart, sustainable and resilient districts. Through the adoption of tools and technologies to support the assessment of actions and policies undertaken by local governments, the project incentivized public space usability and cultural heritage accessibility from a citizen-centred perspective. The deployment of tools and technologies followed the process of research-action-research [51], in which pilot interventions were codesigned with communities, implemented as pilot activities and then continuously monitored with the aim of understanding the impacts and effects.

In the dynamic and changing context of contemporary cities, people's flow is a significant indicator of attractive, accessible and healthy urban spaces [52]. In the city of Bologna, within the framework of the ROCK project, a crowd monitoring system was installed to monitor people flow in some historic central areas in the presence of temporary transformations of public spaces, codesigned and cocreated with citizens and other stakeholders involved in the project.

The aim of this experiment was to use sensor data to assess how the climate and microclimate of urban spaces, such as squares, have influenced people's presence in the area, with and without public events.

Crowd monitoring instrumentation developed by DFRC (Data Fusion Research Centre), a company world leader in Wi-Fi analytics and a partner of the project, has been installed on the Bologna University campus, along Zamboni Street. This LBASense technology, thanks to a network of distributed sensors that count the Wi-Fi or GPS signals generated by smartphones within a radius of 50 m from the sensor, allows for estimating in real-time and in aggregate mode (so as to prevent the identification of passersby) the number of people present in an area, the duration of their stay and their origin country [53].

The collected data are openly accessible and downloadable through the ROCK platform (https://www.rockproject.eu/ accessed 2 May 2023). In the dashboard section of the platform, different query tools allow for dynamically visualising data through different types of charts and diagrams and it is possible to download data in CSV format in relation to the location of sensors, the time of interest, the temperatures recorded, the nationality of visitors and the movements of passengers between different points of interest in the city.

The experiments carried out by the ROCK project allow for validating the importance of providing urban planners and policymakers with tools dedicated to measuring the impact of implemented actions [54], demonstrating how innovative solutions that improve livability, accessibility and resilience in CH contexts can result from an appropriate use of big datasets, now available to an increasing extent.

The scientific literature does not report studies or research that directly correlates the presence of people in outdoor spaces with thermal comfort and outdoor microclimate simulation, specifically using Envi-met software. In fact, from an explorative search carried out in the Scopus database with the keywords "people flow" or "people presence" and "Envi-met", only the following articles were found [35,55]. The search with the keywords "pedestrian comfort" and "Envi-met" reports less than a hundred publications such as [27,56–58], and, similarly, the keywords "people flow" or "people presence" and "outdoor thermal comfort" show the following articles, which are more focused on individual subjects than on people flows [59–61].

In spite of that, some research share some aspects with the present one. A study about thermal comfort in urban space carried out with interviews about people's thermal comfort described by Nikiolopoulou et al. [62], where the authors compare 1431 interviews with on-site measured data with a minimet station and globothermometer; a study by Nikiolopoulou and Lyjoudis [63], who have carried out 1503 interviews correlating them with meteorological parameters and use of space; and other research in the Nordic climate contexts, by Eliasson et al. [64], or in Japan, by Thorsson et al. [65] in urban public places. All these studies adopt interviews to define people's thermal comfort and microclimate data measurement on site. Within this framework, when analysing the relationship between microclimate and people's behaviour, a different approach was adopted. In the case of the present research, people's thermal comfort perception depends on the crowd's behaviour in space, and microclimate data are not directly measured but simulated with Envi-met; this simulation was later validated by in situ climate data monitoring. A similar approach was adopted in the article by Chokhachian, Santucci and Auer [52] in Boston, where the UTCI index was used instead of PET, the GPS-tracked pedestrian activity used in Bologna to measure people's presence. In Boston, the authors did not take into account any specific public events to define the role of microclimate as a determinant for staying in an urban space. This study, therefore, represents an innovative contribution in a little-explored research area in which there are still limited and partial experiences.

The question is whether a direct correlation exists between outdoor microclimate in public space and people's presence, and if a public event plays a role in altering it. This appears quite evident. If open space is too hot, people prefer to stay at home, except in the case of a public event, such as a concert or a show.

Within the research field of urban and comfort studies, the authors propose a methodology to evaluate whether the above observation is true in every case in order to identify health risks during mass gathering events [52–67] and the role of public space in preventing or triggering these health risks. The present research did not propose a survey or

questionnaire to people in order to discover their thermal sensations; instead, a different methodology was adopted, inferring that if people stayed in the space, they showed that they deemed the thermal condition acceptable or comfortable. The main novelty of the study concerns the interrelationship between these three items: thermal comfort in outdoor space, the architecture of public space and social events in the city centre. In our case, as the results show, both public open space and public events have no negative effect on people's presence.

## 2. Goals

The goal of this research is to investigate the relationship between outdoor thermal comfort and people's presence in public open areas.

The outdoor microclimate simulation in urban spaces, with Envi-met software, allows for verifying the relationship between the built environment and natural scenario characteristics, such as urban fabric conformation, building architectural features, materials and vegetation effects and local physical variables (air temperature, relative humidity, surface temperature, wind speed). The characteristics of the modelled scenario and the physical variables can be then related to users' comfort perception expressed through biometeorological models, such as physiological equivalent temperature (PET) or universal thermal climate index (UTCI), which allow for describing the overall thermal sensation of a person standing in the virtual environment of the analysed scenario.

The results of the simulations carried out on the virtual model show all the above-mentioned characteristics for outdoor thermal comfort determination and allow for the description of the environment's physical variables. These results can be compared with the measured in situ real data so as to validate and verify the correspondence between model and reality.

PET evaluation allows us to connect the comfort perceived by people in the built environment with the people flow detected through City People Flow tools (DFRC) and, then, identify the relationships between microclimatic context and people's presence. This reflection will necessarily have to consider that the urban microclimate is the sum of independent meteorological factors that merge and coalesce at the human body as an individual sensation of the local climate conditions and that those conditions depend on urban and architectural characteristics, but also on the choice of materials and vegetation that contribute to urban space definition.

The ROCK project adopted tools and technologies as key elements to deepen the understanding of historic city centres behaviours, with the aim to sustain data-based actions of regeneration and transformation in sustainable and more resilient districts, considering the vulnerabilities and constraints of the specific cultural heritage contexts. This project implemented multiple sets of integrated pilot actions to trigger the reactivation of neglected or underused public spaces in the university area in Bologna. The quality of public spaces is an essential component for the citizen quality of life. Thus, the ROCK project aimed to test actions, strategies and tools to foster the usability and accessibility of public spaces to all, fostering cultural heritage enhancement from a user perspective, integrating physical transformations with new uses, also unconventional, to bring life and social inclusion possibilities for public spaces. The study of the relation between the microclimatic conditions in such public spaces and the presence of people was one of the research inquiries; the use of sensors and assessment tools was at the base of the methodology. In particular, the present paper aims to evaluate whether the presence of people in public squares was due only to the organization of events or it was also related to microclimatic conditions.

In other words, if it is recognised that a public event has a main role in triggering people's presence inside a public space, we aim to evaluate if outdoor microclimate also plays a role, comparing moments with and without public events. In this way, our research allows us to make an innovative contribution to the research field in outdoor microclimate, pedestrian comfort, cities and society.

## 3. Case Study

The research presented in this paper uses the Envi-met software to simulate outdoor microclimate and thermal comfort in two central areas of the city of Bologna, located in the ancient university district. Bologna is located in the north of Italy and has a dense and significant historical city centre (Figure 1). The climate is warm and temperate and it shows significant rainfall, even in the driest months. According to Koppen categories, the climate has been classified as Cfa—humid subtropical climate [68,69]. The average annual temperature in Bologna is 14.3 °C, while 825 mm is the average annual rainfall.

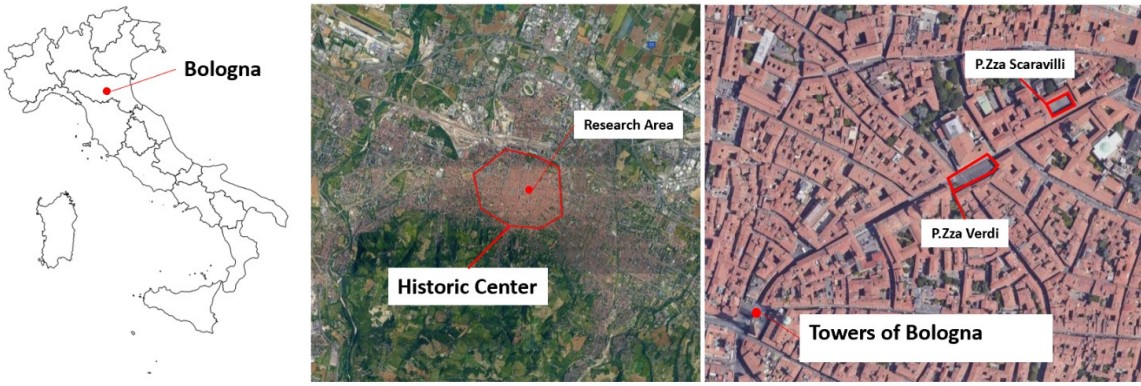

**Figure 1.** Italy, Bologna and historic center of Bologna.

The activity foresees the integration of different tools and sensors to evaluate the spatial distribution of the above factors and obtain a combination of data useful to support strategic actions at the demonstration sites. In fact, the possibility to detect the presence of people in public spaces allows us to rate event successes, the effects of spatial transformations and responses to regeneration strategies in order to be able to design urban strategies and support decision-making addressed to regeneration and re-activation in CH contexts.

In detail, our research reported the correlation between people density (people flow, count and density) and outdoor thermal comfort for Piazza Verdi and Piazza Scaravilli (Figure 2), two of the most important squares along via Zamboni. The two areas are densely built, and the urban fabric is very compact.

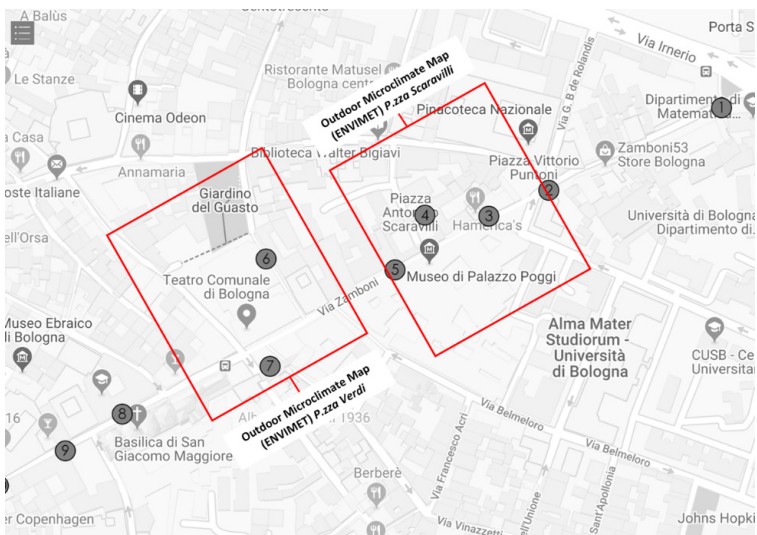

**Figure 2.** Area with DFRC sensors and Envi-met area simulation. Right Piazza Verdi and left Piazza Scaravilli area. The numbers in the image are the numbers of local sensors.

Piazza Verdi's dimensions are 26 m in width and 72 m in length with an area of 1872 m$^2$; buildings average height around the place is 12 m (approximately four floors), except along the north side where the Municipality Theatre of Bologna is located, with an 18-m height. The surrounding buildings do not shade the square. The sky view factor is 0.20. The pavement of the square is granite (single stone) with an albedo equal to 0.4.

Piazza Scaravilli is 50 m wide and 56 m long, with an area of 1120 m$^2$, and it is surrounded by a single 4-m wide and 8-m tall building, with a portico (height 4 m) along all four sides. The other building around the square is 12 m tall. The sky view factor is 0.45. All buildings have brick exposed. The pavement is in granite (single stone), the same as Piazza Verdi, with an albedo equal to 0.4.

Climate data and weather conditions for four days of simulation are reported in Table 1. In any case, wind speed measured at 10 m height is 2.50 m/s wind direction of 90 deg.

**Table 1.** Climate data, air temperature and relative humidity for four simulation days from ARPAER [70].

| Day | Hour | Air Temperature (°C) | Relative Humidity (%) | Day | Hour | Air Temp.(°C) | Relative Humidity (%) |
|---|---|---|---|---|---|---|---|
| 26 June 2019 | 00 | 24.3 | 64 | 16 August 2019 | 00 | 21.3 | 67 |
| | 01 | 23.6 | 66 | | 01 | 21.2 | 60 |
| | 02 | 23.4 | 65 | | 02 | 19.2 | 63 |
| | 03 | 23.1 | 65 | | 03 | 18.8 | 71 |
| | 04 | 22.5 | 66 | | 04 | 18.7 | 77 |
| | 05 | 22.7 | 64 | | 05 | 18.9 | 82 |
| | 06 | 23.8 | 60 | | 06 | 19.8 | 77 |
| | 07 | 26.4 | 51 | | 07 | 20.3 | 74 |
| | 08 | 27.9 | 49 | | 08 | 21.7 | 67 |
| | 09 | 30.2 | 42 | | 09 | 22.6 | 62 |
| | 10 | 31.3 | 39 | | 10 | 24.3 | 52 |
| | 11 | 32.2 | 37 | | 11 | 25.8 | 40 |
| | 12 | 33.0 | 36 | | 12 | 26.5 | 36 |
| | 13 | 33.4 | 33 | | 13 | 27.1 | 35 |
| | 14 | 33.8 | 33 | | 14 | 27.4 | 33 |
| | 15 | 33.9 | 32 | | 15 | 27.7 | 33 |
| | 16 | 33.9 | 31 | | 16 | 27.6 | 32 |
| | 17 | 33.8 | 31 | | 17 | 27.4 | 32 |
| | 18 | 33.0 | 33 | | 18 | 26.7 | 39 |
| | 19 | 32.1 | 38 | | 19 | 25.5 | 44 |
| | 20 | 30.5 | 45 | | 20 | 24.5 | 49 |
| | 21 | 28.1 | 55 | | 21 | 23.7 | 51 |
| | 22 | 27.0 | 61 | | 22 | 22.7 | 55 |
| | 23 | 26.3 | 62 | | 23 | 21.4 | 61 |
| 27 June 2019 | 00 | 25.7 | 62 | 17 August 2019 | 00 | 20.7 | 63 |
| | 01 | 25.4 | 61 | | 01 | 19.9 | 65 |
| | 02 | 25.2 | 61 | | 02 | 19.5 | 65 |
| | 03 | 24.8 | 62 | | 03 | 19.3 | 64 |
| | 04 | 24.6 | 62 | | 04 | 18.9 | 65 |
| | 05 | 24.8 | 61 | | 05 | 18.7 | 65 |
| | 06 | 26.0 | 57 | | 06 | 19.6 | 62 |
| | 07 | 27.9 | 52 | | 07 | 21.9 | 54 |
| | 08 | 30.3 | 45 | | 08 | 23.0 | 50 |
| | 09 | 31.9 | 41 | | 09 | 25.1 | 43 |
| | 10 | 33.1 | 36 | | 10 | 26.4 | 41 |
| | 11 | 34.6 | 32 | | 11 | 27.7 | 35 |
| | 12 | 35.7 | 29 | | 12 | 29.1 | 31 |

**Table 1.** *Cont.*

| Day | Hour | Air Temperature (°C) | Relative Humidity (%) | Day | Hour | Air Temp.(°C) | Relative Humidity (%) |
|---|---|---|---|---|---|---|---|
| | 13 | 36.9 | 25 | | 13 | 29.4 | 29 |
| | 14 | 37.3 | 24 | | 14 | 30.1 | 27 |
| | 15 | 37.7 | 20 | | 15 | 30.2 | 28 |
| | 16 | 37.7 | 22 | | 16 | 30.1 | 28 |
| | 17 | 37.8 | 22 | | 17 | 29.4 | 34 |
| | 18 | 37.4 | 25 | | 18 | 28.2 | 41 |
| | 19 | 35.4 | 34 | | 19 | 26.9 | 46 |
| | 20 | 32.1 | 45 | | 20 | 25.9 | 49 |
| | 21 | 30.7 | 52 | | 21 | 25.2 | 51 |
| | 22 | 30.4 | 49 | | 22 | 23.9 | 56 |
| | 23 | 29.9 | 47 | | 23 | 23.1 | 57 |

## 4. Methodology

The aim of the research is to evaluate the relationship between: (i) physics and climate data of an open space (e.g., Piazza Verdi and Piazza Scaravilli in Bologna) and (ii) people flows (how many people). Human presence and fruition of open space depend on several factors, which our research aims to integrate. Among them:

1. outdoor microclimate factors, described by physical variables (e.g., air temperature, relative humidity) and thermal comfort by people (physiological equivalent temperature, PET);
2. effects of cultural or social events, initiatives, presence of specific cultural and commercial activities that could have an influence on people flows and grouping.

In our specific case, measurements were conducted before and after having implemented ROCK actions (temporary transformations, events and initiatives). Three types of data were adopted:

(a) Outdoor physics variables combined with real data measured on site;
(b) Outdoor Microclimate Maps (OMM) as defined in Gaspari and Fabbri [4,71] obtained by virtual simulations by the software Envi-met [8] that allow to measure outdoor thermal comfort;
(c) Measurement of people flow by hours and other data available from the LBASense system.

The physical parameters detected on site (a), in particular air temperature, are used for the calibration of simulations with Envi-met (b), as described in Section 5.1.

The research consists of OMMs comparison at different times of the day with related people crowding on days both with and without events. Figure 3 shows a graphical abstract of our methodology.

### 4.1. Outdoor Physics Variables

Outdoor physics variables were detected through the use of sensors deployed by Acciona, a technological partner of the project (https://www.acciona.com/projects/rock/ ?_adin=02021864894, accessed on 2 May 2023). Through ICT sensors and tools, ACCIONA supported the concrete and on field application of the ROCK project strategy. A monitoring tool was set up from the very beginning and will probably be running for two additional years after the project's lifetime. The environmental parameters measured on site by sensors are: air temperature (°C), absolute and relative humidity (%), air pressure (Pa) and PM10 (ppm). The parameters were compared and calibrated with the environmental quality office using ARPAER (Agenzia Prevenzione Ambiente Energia Emilia Romagna, Bologna, Italy) data [70]. Figure 4 shows ACCIONA sensors positions in the two study areas.

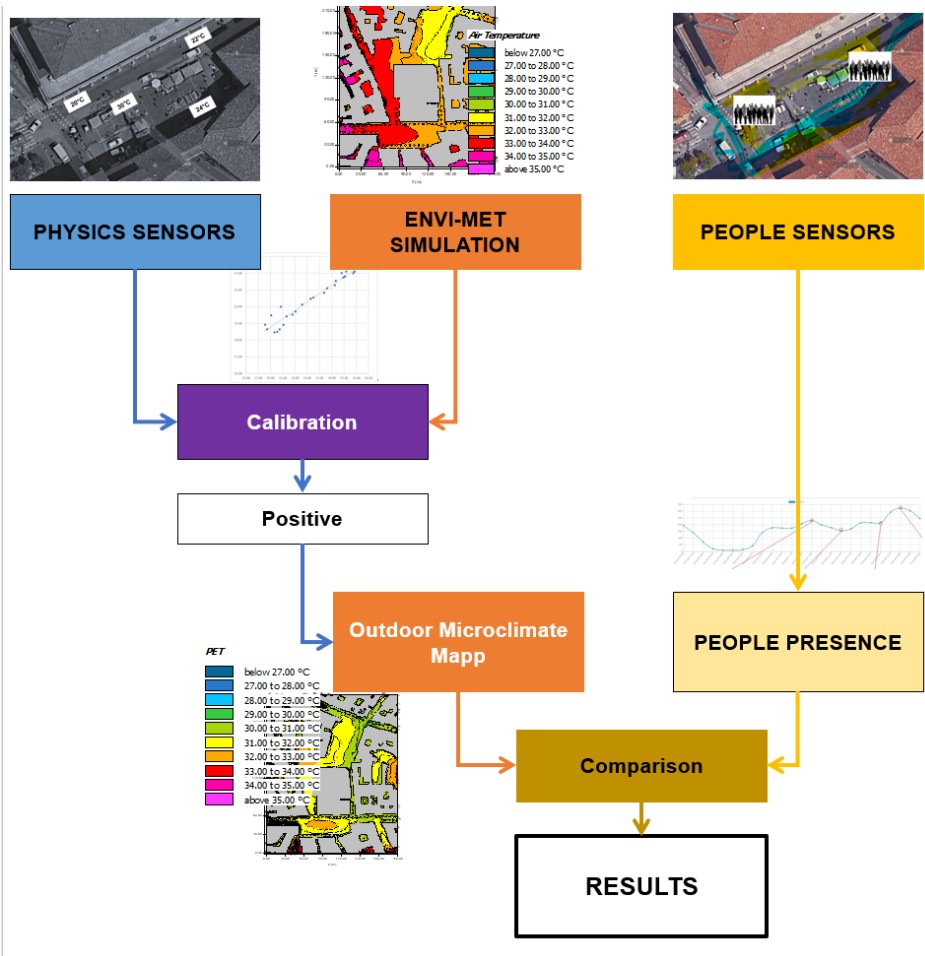

**Figure 3.** Methodology graphical abstract.

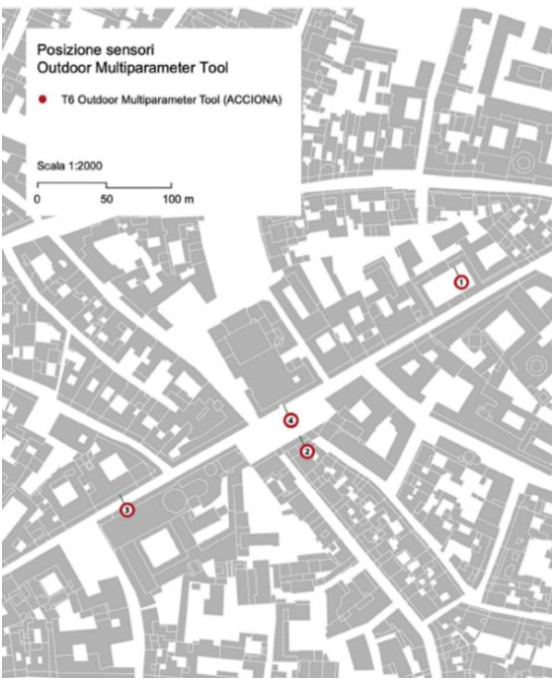

**Figure 4.** Location of ACCIONA outdoor sensors in Via Zamboni—Bologna. Sensors used for calibration are 1 and 4.

Sensors are located at nearly 4–5 m; Table 2 reports the main sensor features.

**Table 2.** Outdoor sensor (data every quarter of an hour).

| Description | Units | Range | Assurance | Resolution |
|---|---|---|---|---|
| Ambient temperature | °C | −40°C to +70°C | ±0.3 °C @20 °C | 0.1 |
| Ambient Relative Humidity | % | 0–100% | ±2% @20 °C (10%−90$ RH) | 1% |
| Atmospheric pressure | Pascal | 300 to 1100 hpa | ±0.5 hPa @25 °C | 0.1 hPs |
| Wind speed | m/s | 0.01 m/s to 60 m/s | ±3% to 40 m/s \| ± 5% to 60 m/s | 0.01 m/s |
| Wind direction | deg | 0–359 °C | ±3% to 40 m/s \| ± 5% to 60 m/s | 1° |

The ACCIONA ROCK monitoring system is a modular platform for remote monitoring, follow-up and analysis of any type of environmental parameter. An interoperable platform enables the collection and exchange of data. The platform consists of three differentiated layers: monitoring hardware, data collection and transmission and a web interface.

The monitoring hardware layer comprises all the physical devices (sensors, power supply, communications, etc.) needed for measuring all parameters that need to be monitored. The data collection and transmission layer acts as a bridge between the hardware monitoring layer of a specific monitoring installation, and the ROCK central remote server that stores the monitored data in each monitoring installation. The collected data and the transmission layer sent to the remote ROCK server will benefit from the most optimal Internet connection available in each ROCK pilot.

*4.2. Envi-Met Simulation and OMM*

Each case study environment was modelled by Envi-Space and Envi-Sim, by using the following model dimension: 210 m × 255 m, equal to 140 × 170 cells. Each cell has a dimension of 1.5 m × 1.5 m, allowing a good detailed representation for each area. At the end of the simulation, the following actions were completed:

- verification of the completeness of outputs;
- simulation, for each scenario, of the physiologically equivalent temperature (PET) with BIO-met;
- creation of an outdoor microclimate map (OMM) with Leonardo 4.4.0 Envi-met.

  An OMM at times 10:00, 12:00 and 14:00 e 16:00 was extrapolated for the following variables:

- air temperature, measured in °C;
- relative humidity, measured in %;
- wind speed, direction and intensity, measured in m/s;
- surface temperature, measured in °C;
- thermal comfort with PET.

The outdoor microclimate map (OMM), obtained through Envi-met, reports two variables: air temperature (°C) and physiological equivalent temperature (PET, measured in °C). Table 3 reports PET values and thermal comfort ranges.

Envi-met simulations were performed for two consecutive days, both in June and August. Each couple of days included a day with a public event (concert or opera drama) and one without public events.

**Table 3.** Physiological equivalent temperature (PET, in °C) from [72].

| PET (°C) | Thermal Perception | Thermal Stress |
|---|---|---|
| 18 | Comfortable | No thermal stress |
| 23 | Slightly warm | Slightly thermal stress |
| 29 | Warm | Moderate thermal stress |
| 35 | Hot | Strong thermal stress |
| 41 | Very hot | Extreme thermal stress |
| >42 | Too hot | Very extreme thermal stress—heat stroke risk |

*4.3. People Flow Measurement by DRFC*

The crowd monitoring instrumentation, developed by DFRC and used within the ROCK project, uses the LBASense system, a network of sensors (Wi-Fi/GPS—4 kits), placed in the demonstration areas. The LBASense system analyses data over time and provides real-time insights into activity and mobility patterns within the monitored area. This combination of sensors allows the system to perform a more comprehensive reading of the crowd's nature, enabling end-users to access demographic figures and mobility patterns on a wider scale, in ordinary and extraordinary conditions (i.e., festivals and events). The goal is to determine footfall in real-time and change rate in the crowd size, besides dwell time, revisit and flow patterns and distribution, as well as any abnormalities, in the overall area under monitoring, with an additional focus on selected key locations.

The pedestrian crossing detection sensors are inserted in 24 × 19 × 9 cm derivation boxes and installed next to the existing derivation boxes along with video surveillance cameras of the Bologna Municipality (Figure 5).

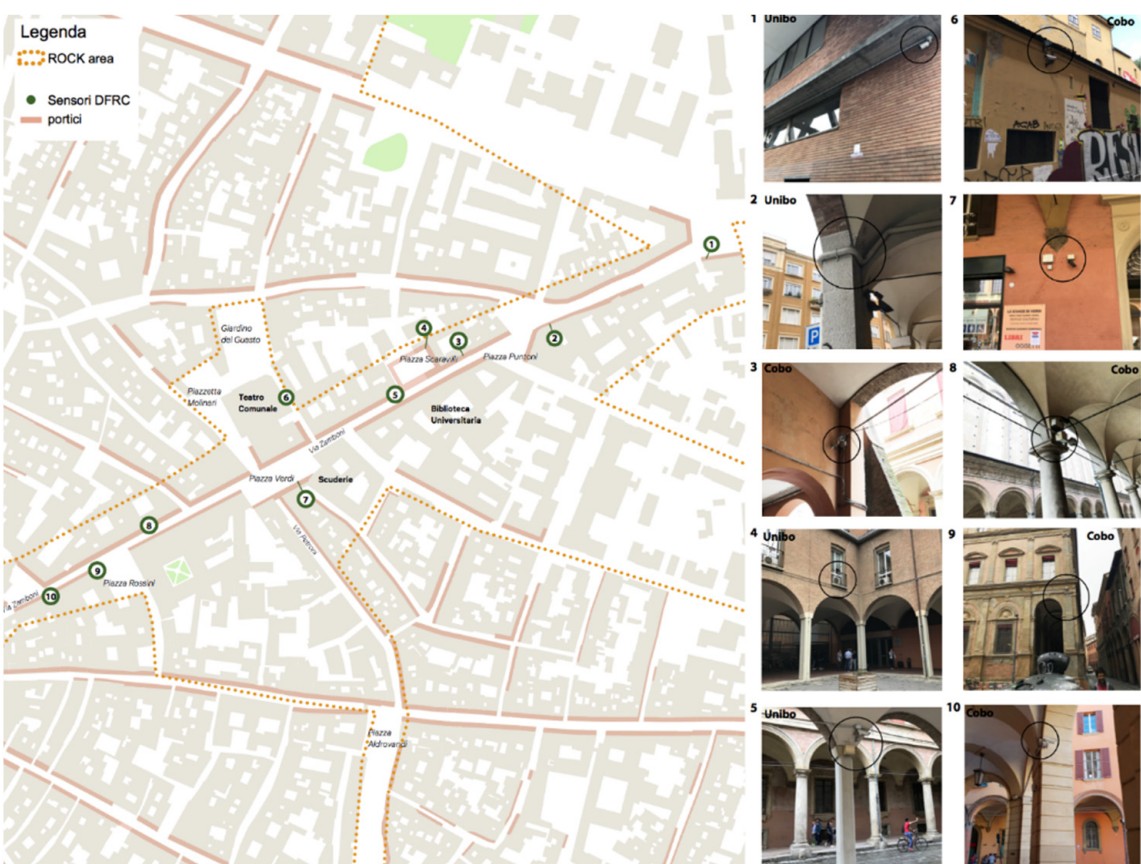

**Figure 5.** Position of DFRC sensors in the area. The number in the image are local sensors.

DFRC sensors measure crowds of people (giving aggregate and anonymous data) thanks to smartphone Wi-Fi and/or GPS signals. The data collected concerns the number of people present in an area (approximately 50 m radius from the sensor), their movements within the covered area and their nationalities. The accuracy compared to the actual presence has been calculated at approximately 91%, comparable to the one of much more expensive technologies that make use of cameras. The data are anonymized and processed in an aggregated way in order to be returned on the ROCK project platform or on detailed platforms, as well as through mobile applications.

We obtained data for Piazza Verdi and Piazza Scaravilli, from the website provided by DFRC: "LBA sense Proactive Business Intelligence" "Visitor Count" from sensors "7—Piazza Verdi", "6—via Del Guasto", "5—Palazzo Poggi", "4—Piazza Scaravilli". Figure 6 reports an example of these data.

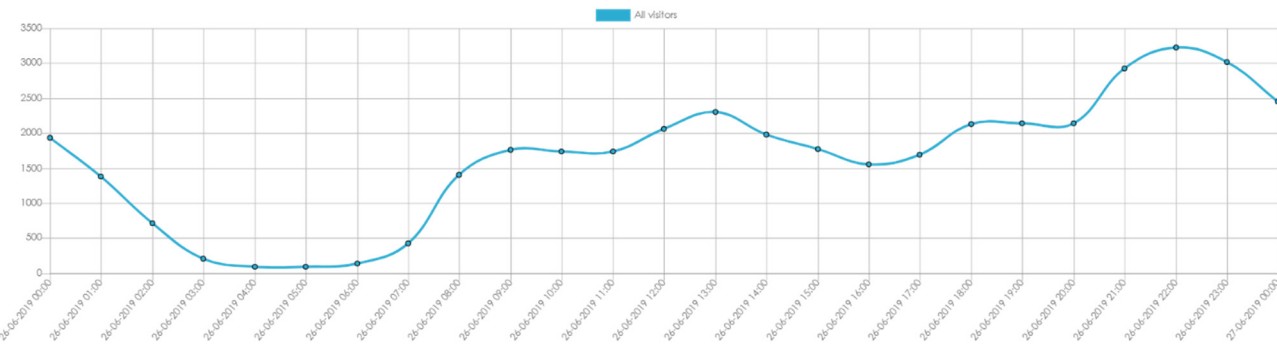

**Figure 6.** Visitor count example. Output by DFRC (x-axis hours, y-axis: number of people).

Data collected by DFRC sensors are people counting (i.e., No. of people detected), on a daily, hourly and per minute resolution, per region and for the entire site; and visit duration distribution (i.e., no. of people per visit's duration range), on a daily and hourly resolution, per region and for the entire site; for the research reported in this article, only people counting is relevant.

Data are available for visualisation and monitoring via a credential-protected dashboard (web-based and mobile app), and they are integrated via APIs (application programming interfaces).

Users who are given an account to access data, upon the municipality's request, can select and export data per region and time span by any of the above-mentioned means; for some queries, like real-time counting, there are constraints related to data availability that prevent the system from crashing due to workload (e.g., via APIs, real-time data ca nnot be retrieved for dates earlier than the day before the current one; via dashboard, real-time data is available in read-only mode).

## 5. Simulations and Results

The results were processed at the end of the monitoring campaign and simulations with Envi-met. The first step was the Envi-met results calibration necessary to correctly consider the OMMs. The validation of the Envi-met model was carried out following previous studies [25,44,45]. Once it was confirmed that the OMMs were calibrated according to on-site measurements of environmental parameters, the extrapolation of the OMMs related to air temperature and PET was performed, as described below.

### 5.1. Envi-Met Results Calibration

In order to calibrate the Envi-met model, a test simulation on 10 January 2019 for each selected area was performed, adopting the same outdoor variables and data (air temperature, relative humidity, wind speed, etc.) recorded in the city of Bologna (DEXTER—ARPER source and Acciona on-site measurement data). In this first step the aim was only

to check the model (adequate size, data, vegetation, etc). Each simulation required between 110 and 130 h of simulation running. As shown in the following paragraph, air temperature was used to validate the model with measured data and the input data based on the preliminary results were not needed.

In this research, on-site climate data were monitored through ACCIONA sensors located in Piazza Verdi and Piazza Scaravilli. In order to calibrate the model with the measured data, air temperature (°C) was used, as usual in scientific literature, as it is not so influenced by local meteorological factors such as wind, rain, etc.

Both Piazza Verdi and Piazza Scaravilli models were calibrated considering the 26 June and the 16 August 2019: being their results homogenous, it can be ensured that all models are calibrated, without significant errors.

Figure 4 shows the location of ACCIONA sensors in the area. For the research scopes, sensor codified 102 for Piazza Verdi calibration and sensor 101 for Piazza Scaravilli were used.

The calibration concerns a statistical comparison between on-site measured data, in this case through Acciona sensors, and Envi-met simulation data, both referring to the specific points where sensors are located. In other words, real and simulated data of air temperature were compared at the same point, following the ASHRAE Guideline 14-2014, "*Measurement of Energy, Demand, and Water Savings*" which explains a calibration statistical index.

Tables 4–7 reports the statistical indexes: mean bias error (MBE, model calibrated if result less than 10%); root-mean-square error (CV RMSE, results calibrated if is less than 30%); Pearson coefficient (if there is a correlation between 0.3 and 0.7, if greater than 0.7 is a strong correlation); and linear regression $R^2$ (if there is a correlation between 0.5 and 0.7, if greater than 0.7 is a strong correlation).

**Table 4.** Piazza Verdi 26 June 2019—calibration results.

| Index | Results | Calibration Results | ASHRAE Guideline 14 Value |
|---|---|---|---|
| MBE [%] | 6.12% | Calibrated | Fall If MBE > 10% |
| CV (RMSE) [%] | 11.70% | Calibrated | fall if MBE > 30% |
| PEARSON | 0.97 | Strong Correlation | >0.7 (strong) \| 0.3–0.7 (correlation) |
| Linear Regression $R^2$ | 0.9363 | Strong Correlation | >0.7 (strong) \| 0.3–0.7 (correlation) |

**Table 5.** Piazza Verdi on 16 August 2019—calibration results.

| Index | Results | Calibration Results | ASHRAE Guideline 14 Value |
|---|---|---|---|
| MBE [%] | 6.12% | Calibrated | fall if MBE > 10% |
| CV (RMSE) [%] | 11.70% | Calibrated | fall if MBE > 30% |
| PEARSON | 0.97 | Strong Correlation | >0.7 (strong) \| 0.3–0.7 (correlation) |
| Linear Regression $R^2$ | 0.9363 | Strong Correlation | >0.7 (strong) \| 0.3–0.7 (correlation) |

**Table 6.** Piazza Scaravilli on 26 June 2019—Calibration results.

| Index | Results | Calibration Results | ASHRAE Guideline 14 Value |
|---|---|---|---|
| MBE [%] | 5.05% | Calibrated | fall if MBE > 10% |
| CV (RMSE) [%] | 8.16% | Calibrated | fall if MBE > 30% |
| PEARSON | 0.80 | Strong Correlation | >0.7 (strong) \| 0.3–0.7 (correlation) |
| Linear Regression $R^2$ | 0.6459 | Correlation | >0.7 (strong) \| 0.3–0.7 (correlation) |

**Table 7.** Piazza Scaravilli 16 August 2019—calibration results.

| Index | Results | Calibration Results | ASHRAE Guideline 14 Value |
|---|---|---|---|
| MBE [%] | 5.43% | Calibrated | fall if MBE > 10% |
| CV (RMSE) [%] | 8.82% | Calibrated | fall if MBE > 30% |
| PEARSON | 0.78 | Strong Correlation | >0.7 (strong) \| 0.3–0.7 (correlation) |
| Linear Regression $R^2$ | 0.6048 | Strong Correlation | >0.7 (strong) \| 0.3–0.7 (correlation) |

Figures 7–10 report graphics of air temperature as measured by ACCIONA sensors and through Envi-met simulations, and linear regression $R^2$ graphics.

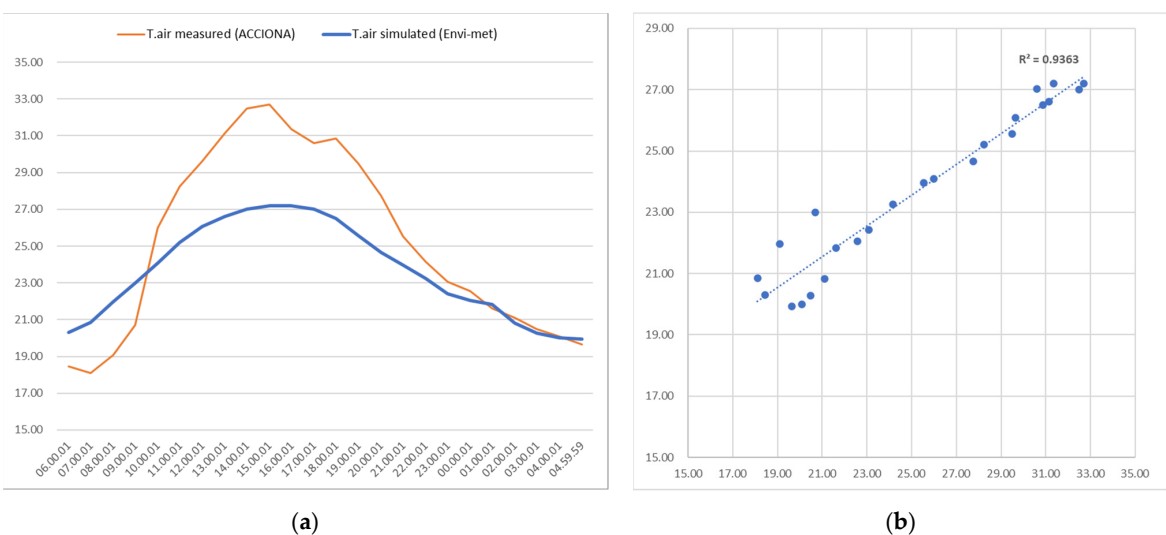

(**a**)  (**b**)

**Figure 7.** Piazza Verdi on 26 June 2019. (**a**) Trend air temperature (°C) (x-axis: hours, y-axis: air temperature); (**b**) graphics linear regression $R^2$ (x-axis: measured air temperature, y-axis: simulation air temperature).

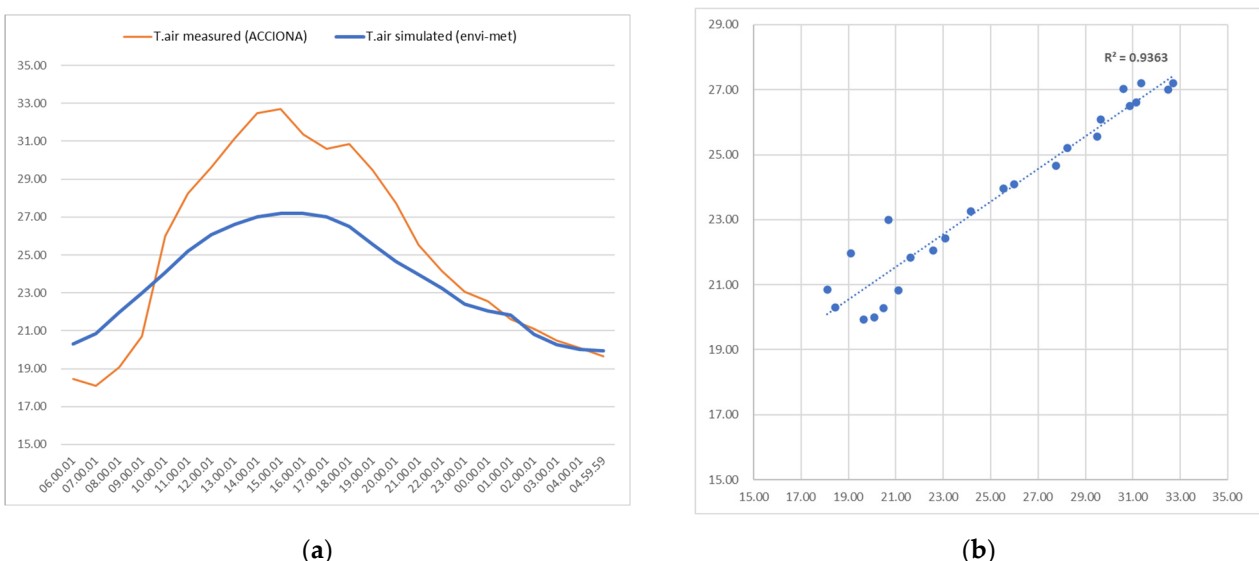

(**a**)  (**b**)

**Figure 8.** Piazza Verdi on 16 August 2019. (**a**) Trend air temperature (°C) (x-axis: hours, y-axis: air temperature); (**b**) graphics linear regression $R^2$ (x-axis: measured air temperature, y-axis: simulation air temperature).

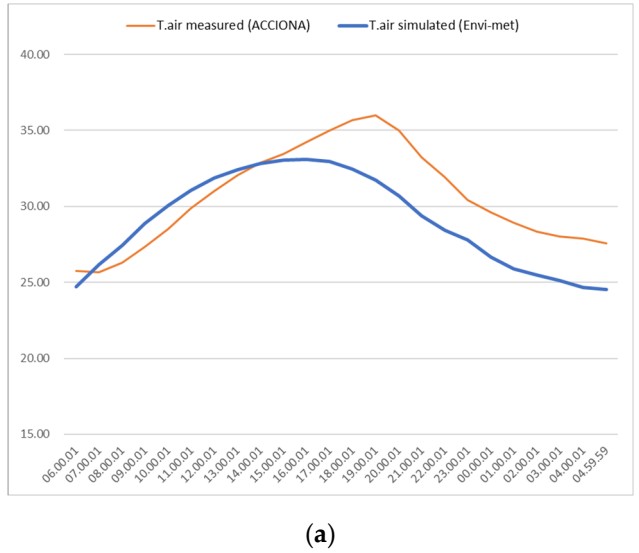
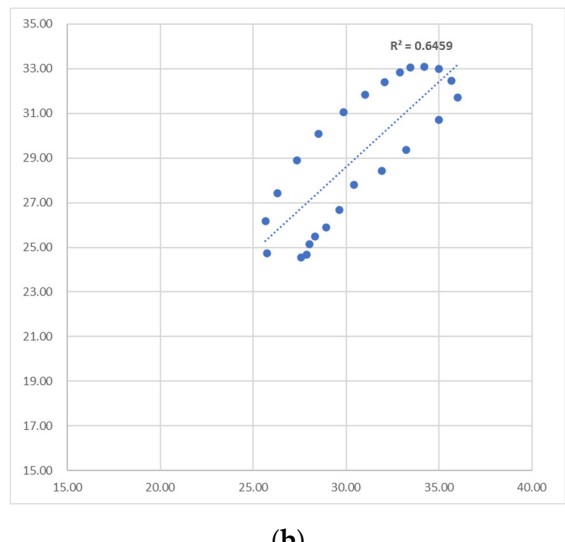

(**a**)  (**b**)

**Figure 9.** Piazza Scaravilli 26 June 2019. (**a**) Trend air temperature (°C) (x-axis: hours, y-axis: air temperature); (**b**) graphics linear regression $R^2$ (x-axis: measured air temperature, y-axis: simulation air temperature).

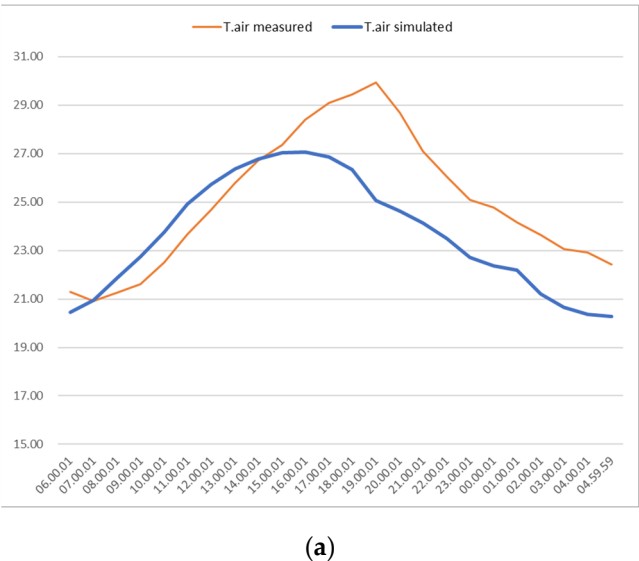
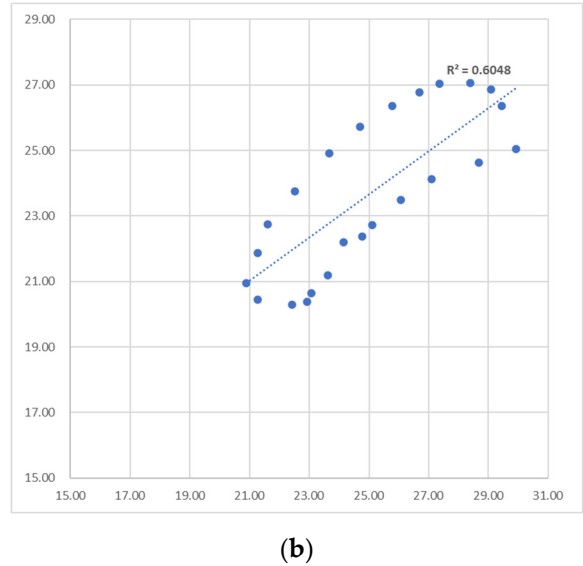

(**a**)  (**b**)

**Figure 10.** Piazza Scaravilli on 16 August 2019. (**a**) Trend air temperature (°C) (x-axis: hours, y-axis: air temperature); (**b**) graphics linear regression $R^2$ (x-axis: measured air temperature, y-axis: simulation air temperature).

*5.2. Envi-Met Simulation Results*

The data related to weather were taken from Dexter ARPAER, the Bologna City weather station. The days of simulations were selected as follows:

- 26 June, during the concert of Neri Marcorè. On this day, the outdoor average air temperature was 27.8 °C;
- 27 June, without any public events and with an outdoor average air temperature of 27.8 °C;
- 16 August, without public events and with an outdoor average air temperature of 23.4 °C;
- 17 August, during the public event Opera Tosca, with an outdoor average air temperature of 24.6 °C.

For each above-mentioned day, the initial boundary conditions were defined as follows:

- starting date: (e.g., 16.08.2017)
- starting time: 06:00
- simulation time: 24 h, with the 2nd day used for evaluation
- wind speed measured at 2 m of height (in m/s): 1.86 m/s
- wind direction (in degrees): E 97°

The duration of the software calculation time was approximately 180–200 h for each simulation.

Four outdoor microclimate maps were simulated for both areas, one for each of the four days, including the variables of air temperature and PET at 12:00, 16:00, 20:00 and 22:00. Sixty-four outdoor microclimate maps were obtained in total.

Figure 11 reports an example of the results' comparison for each dataset, with and without public events, with OMMs at 12:00, 16:00, 20:00 and 22:00 h, and considering air temperature and PET; therefore, it is possible to observe the outdoor microclimate variation and the extension of the crowd for each hour.

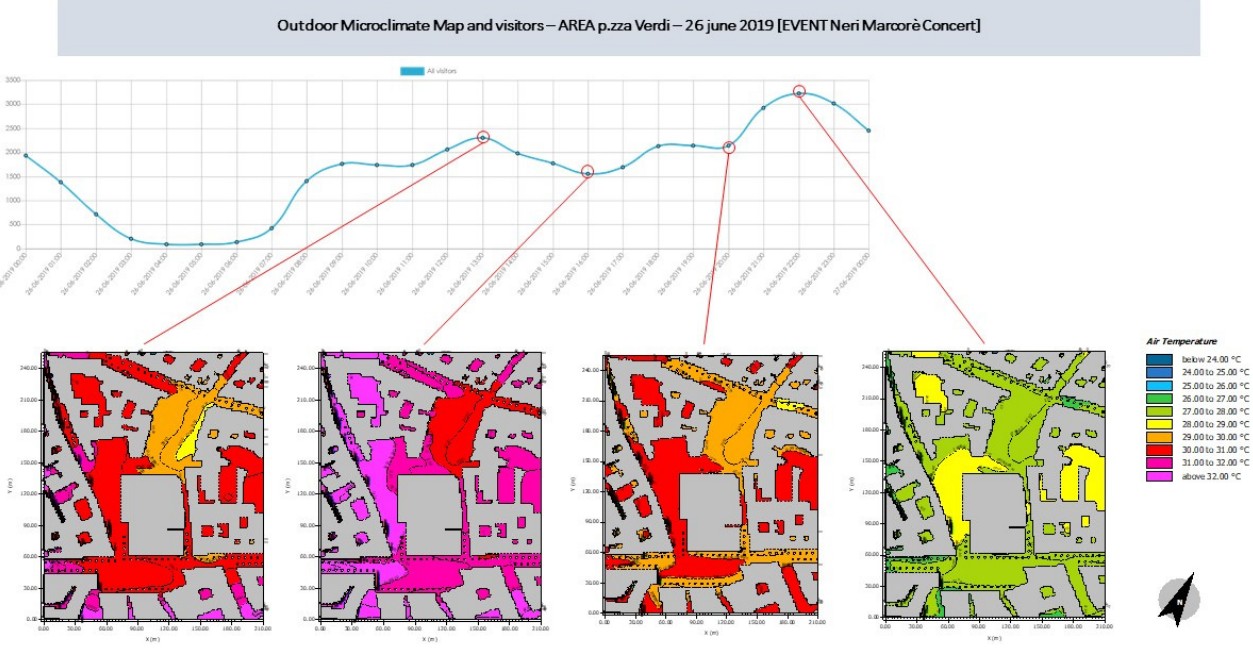

**Figure 11.** Piazza Verdi 26 June outdoor air temperature (°C).

## 5.3. Correlation between Outdoor Microclimate and Visitors Count

Piazza VerdiJune 26 and 27 June 2019, during the concert of Neri Marcorè on 26 June. The OMM comparison (Figures 11 and 12) shows that on the outdoor microclimate, there are some differences between the two days in respect of the air temperature: the 27th of June air temperature is +3 °C higher than the 26th of June one. The air temperature range (or temperature excursion) on those days differs from 32 °C at 12:00, to 27 °C at 22:00. The air temperature distribution in the area is uniform, except in Piazza Verdi where it is slightly lower (−2°C) than in the rest of the area. The visitor count comparison shows (Figure 11) that, at 22:00, the number of visitors was near 3500 people, on the 26th of June, during the public event, while it was near 1500 visitors on the 27th of June. Accordingly, the high number of visitors depends on the presence of the public event and, in part, on the air temperature and outdoor thermal comfort.

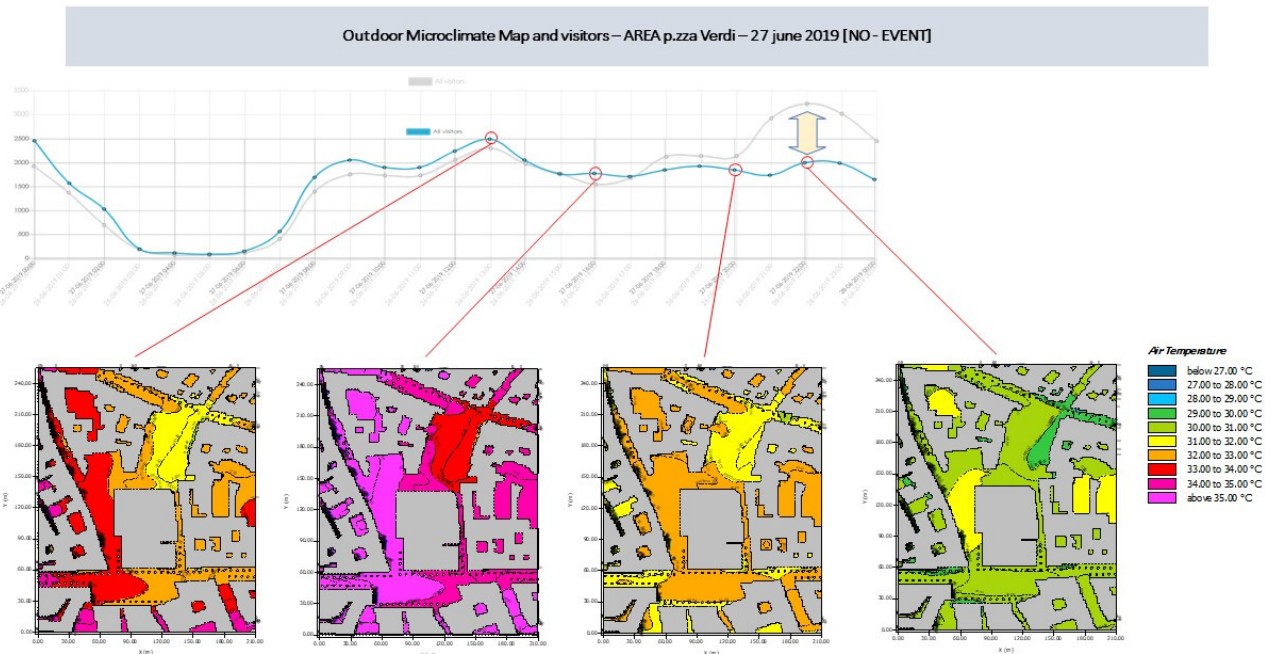

**Figure 12.** Piazza Verdi 27 June outdoor air temperature (°C).

In this case, a gap of 1500 people on site can be observed between the 26th and 27th of June (with and without the event). It is possible to suppose that this gap depends both on events and/or outdoor microclimate. In detail, Table 8 shows that nearly 2000 people stay in Piazza Verdi with 30 °C of air temperature, probably the gap of 1500 people depend sonly by public event.

**Table 8.** Piazza Verdi comparison of the number of people present in the square on 26th June and 27th June in relation to simulation results of air temperature and PET for each day.

|  | 26 June | | 27 June | |
| --- | --- | --- | --- | --- |
|  | 20.00 h | 22.00 h | 20.00 h | 22.00 h |
| People (nearly) | 2000 | 3500 | 1900 | 2000 |
| Air Temperature | 30–32 °C | 26−27 °C | 32–33 °C | 29–30 °C |
| PET | 25°C | <24 °C | 28 °C | <27 °C |

The comparison of thermal perception (PET, Figures 12 and 13) shows a PET value of 24 °C during public events, corresponding to the thermal sensation "slightly warm", while on the 27th of June the PET value is 27 °C, corresponding to the thermal sensation "warm", which produces moderate thermal stress.

Finally, the outdoor microclimate of 26 June (Figure 13) presents two positive conditions: PET values correspond to slightly thermal stress during public events. On the other side, on 27 June (Figure 14) at 20:00 and at 22:00 there was moderate thermal stress without events. This can explain the presence of fewer visitors.

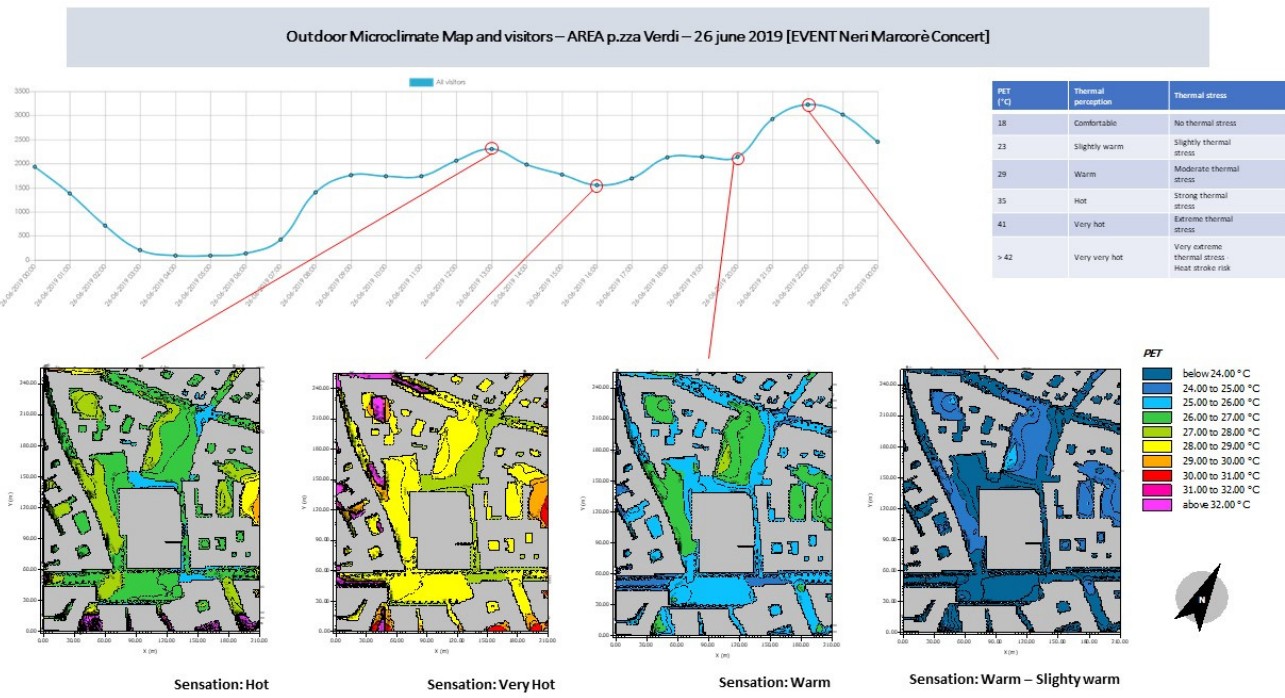

**Figure 13.** Piazza Verdi 26 June outdoor PET (°C).

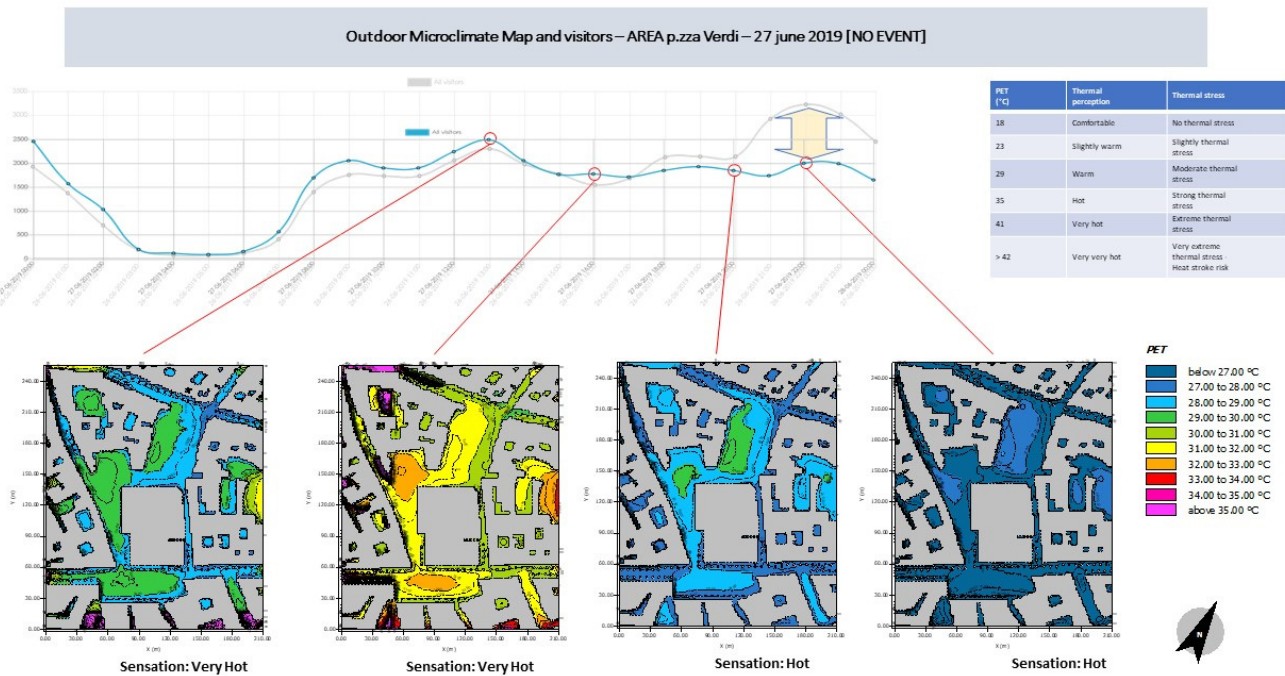

**Figure 14.** Piazza Verdi 27 June outdoor PET (°C).

The comparison of Piazza Scaravilli data (Figures 15–18) during the same days, shows the same increase (+3°C) in air temperature as June 27th, with a uniform air temperature distribution, especially at16:00 and 20:00. The number of visitors are the same for both days, approximately 800–1100 visitors. The PET results need more attention, because the PET records great differences during the same day, at 12:00 and 16:00, with respect to the evening, at 20:00 and 22:00. Thus, two kinds of OMMs with two PET scales were extrapolated. The average value of PET at 12:00 and 16:00 is approximately 36–44 °C which

corresponds to "Hot" e "Very Hot" thermal sensation, with "Strong" and "Very Extreme" thermal stress.

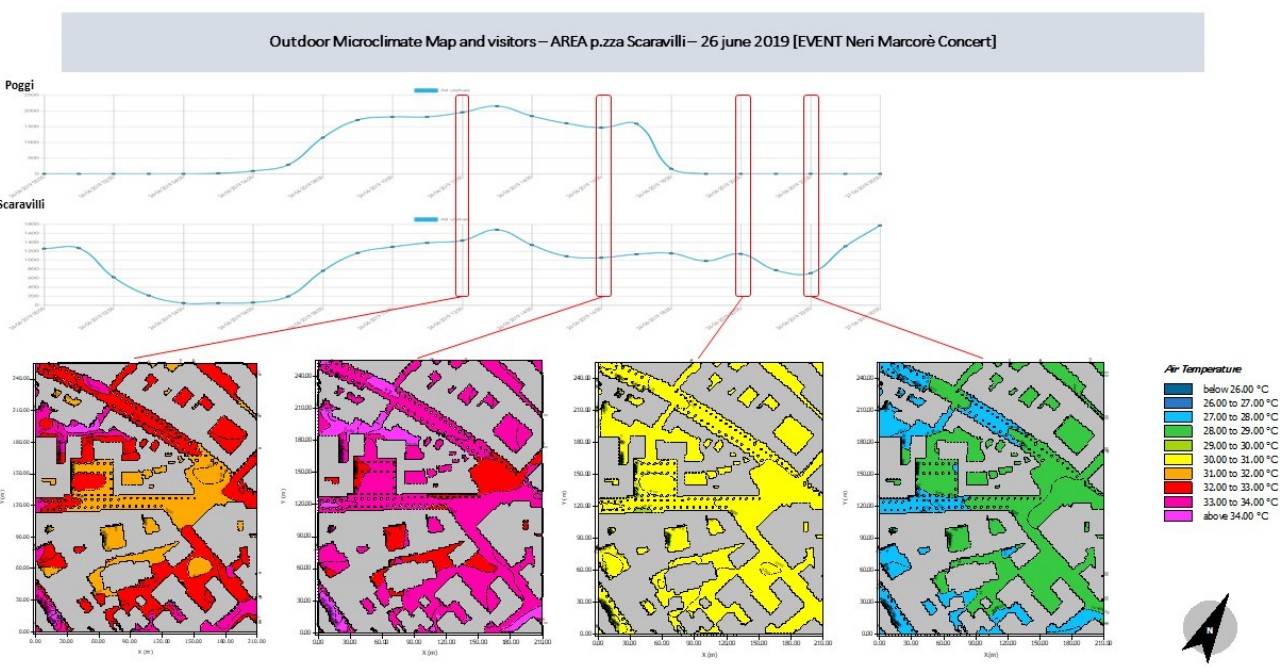

**Figure 15.** Piazza Scaravilli 26 June air temperature (°C).

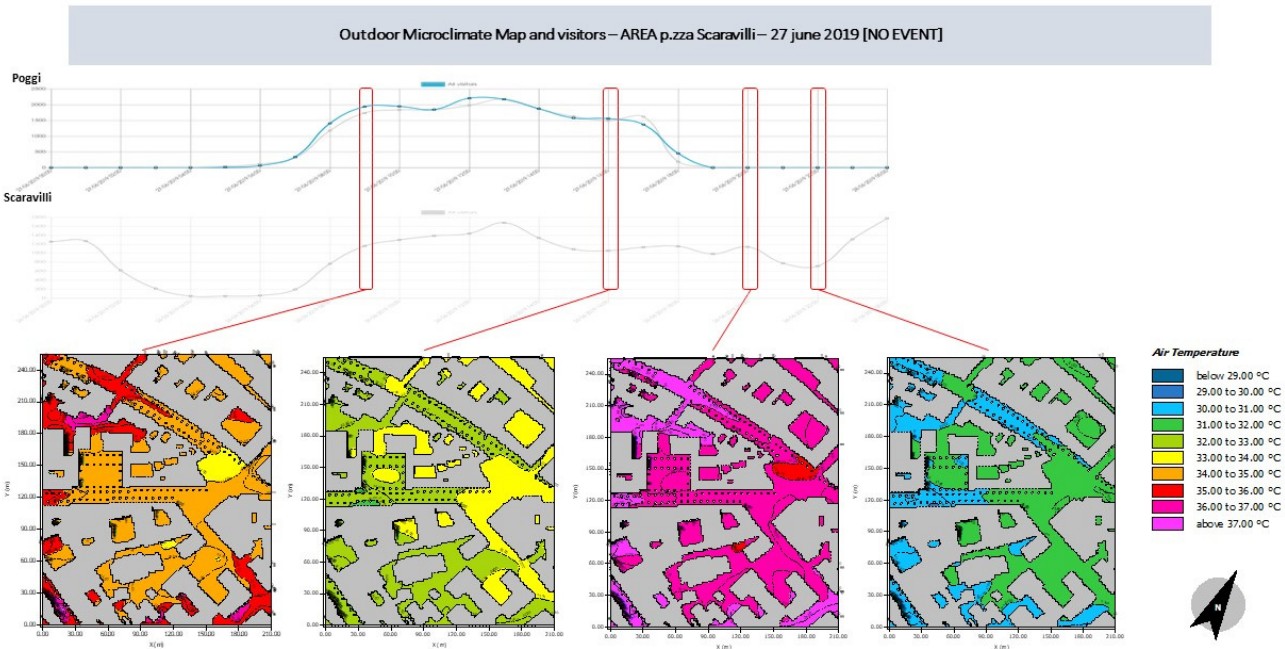

**Figure 16.** Piazza Scaravilli 27 June air temperature (°C).

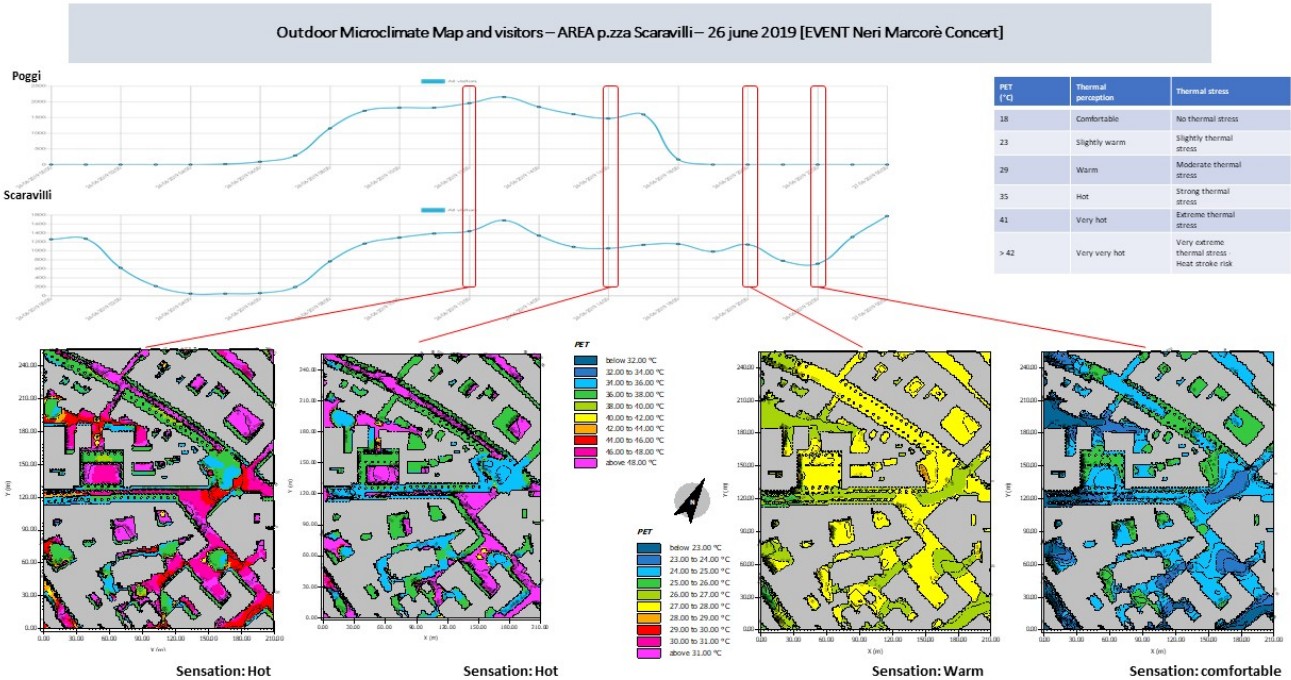

**Figure 17.** Piazza Scaravilli 26 June PET (°C).

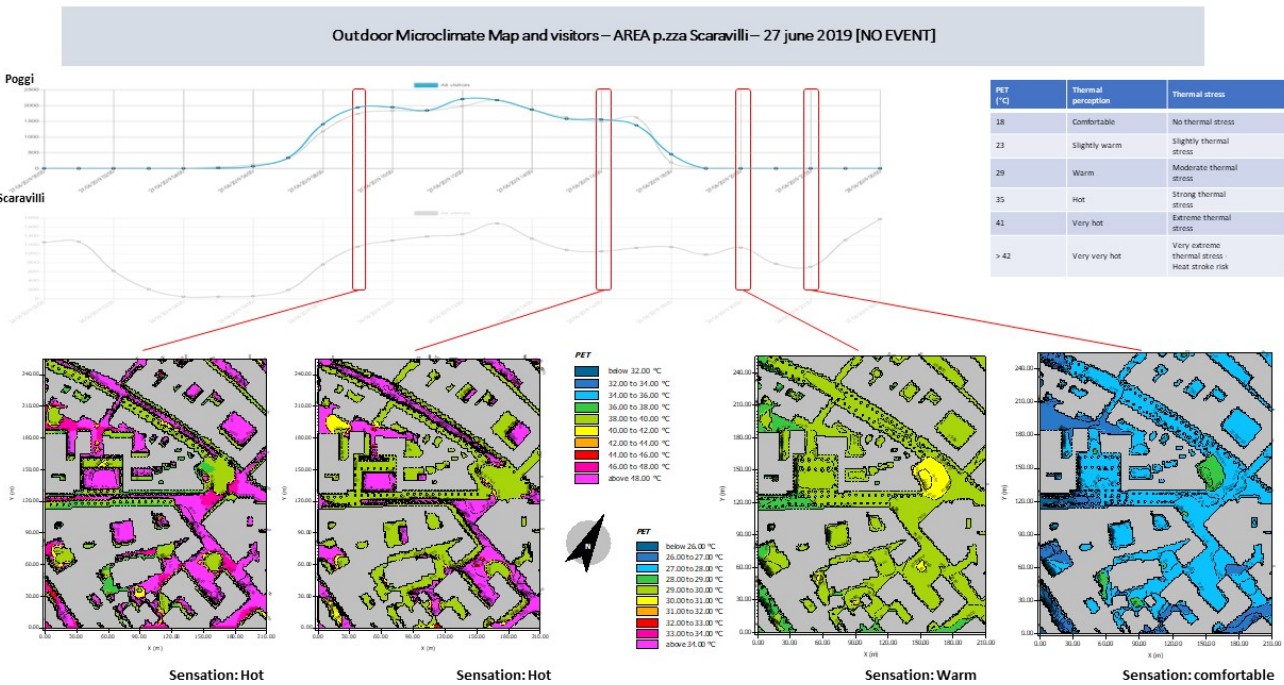

**Figure 18.** Piazza Scaravilli 27 June PET (°C).

The same difference cannot be found in the area of Piazza Verdi. Therefore, this air temperature and PET difference must depend on the specificities of architectural and urban design: building geometry, square geometry, urban canyon, pavement textures, absence of tree or grass, albedo and reflectance of materials.

The above figures show the results of the simulations during 17 August and, the latter during a public event (Opera Tosca). These results confirm the previous discussion but with some specific differences.

Visitors in Piazza Verdi increased from 800 at 22:00 on the 16 August to 1400 on the 17th of August. That depends probably only on the presence of the public event, despite an increase of +1 °C in the air temperature. In both days, at 22:00, the thermal sensation was Comfortable, corresponding to a 20–21 °C PET value.

In Piazza Scaravilli, the PET value was approximately 20–22 °C during the evening, while it was more than 35 °C during the day. In Piazza Verdi the PET value was equal to 24 °C during day and 20 °C during evening.

These data confirm two results: (a) in Piazza Verdi the weather has a high influence on the microclimate but (b) architectural pattern and urban design, especially materials, have a role on outdoor thermal comfort.

The comparison between PET values and number of visitors during the days without events (Table 9, 27 June and 16 August) shows that the numbers of visitors are not directly correlated with the outdoor microclimate conditions, but, probably, they depend on points of attractions (bar, pubs, restaurants, etc.). However, it is possible to argue that PET values corresponding to more comfortable situations (20–21 °C) are often linked with a higher number of persons outdoor, while in presence of less comfortable thermal sensations, e.g., in Piazza Verdi, the number of visitors depends more on public events (Table 10, 26 June and 17 August).

**Table 9.** Comparison of the number of people presents in both squares in relation to simulation results of PET and Visitors Count at 22:00 h, both areas WITHOUT public events.

| Area | Date | PET (°C) | Persons (Num.) |
|---|---|---|---|
| Piazza Verdi | 27th of June | 27–28 | 1500 |
| | 16th of August | 20–21 | 900 |
| Piazza Scaravilli | 27th of June | 21–22 | 600 |
| | 16th of August | 20–21 | 200 |

**Table 10.** Comparison of the number of people presents in both squares in relation to simulation results of PET and Visitors Count at 22:00 h, both areas WITH public events.

| Area | Date | PET (°C) | Persons (Num.) |
|---|---|---|---|
| Piazza Verdi | 26th of June | 24–25 | 3500 |
| | 17th of August | 20–21 | 1400 |
| Piazza Scaravilli | 26th of June | 24–25 | 600 |
| | 17th of August | 20–21 | 250 |

## 6. Discussion and Limits

Research results show that the comparison of homogeneous physical areas simulated through Envi-met to obtain outdoor thermal comfort maps (by PET) can be correlated with outdoor comfort and the number of visitors. Moreover, the proposed methodology allows for a first method to evaluate how architecture and urban design impact thermal comfort and, consequently, crowds and visitors' presence in a public space. However, it is possible to say that more aspects and indicators can influence people's presence in urban space, even if microclimate conditions seem to be important. Nevertheless, microclimate is influenced by several aspects. Thus, this paragraph includes a general discussion of the results and the identification of the major limitations of the research.

The results can lead to reflection by comparing the number of visitors, for each hour, the outdoor microclimate maps, and the possible impacts on public events on the microclimate conditions of the area. These hypotheses were followed: (a) DFRC sensors show the number of persons in the area, so if the number of visitors is the same with/without events, it can be supposed that the crowd does not depend on the presence of events but only on

outdoor microclimate; (b) if the visitor count graph reports a visitor count difference with the same outdoor microclimate conditions, it is possible to argue that the crowd depends mainly on public events.

Results demonstrate that air temperature and thermal comfort (PET) in outdoor spaces depend on the specific characteristics of the urban space, especially considering its geometry, the building fronts and their shadow on the open space, and thermophysics parameters such as reflectance and/or albedo.

In fact, Piazza Verdi is surrounded by tall buildings: the large volume of the Theatre (Teatro Comunale), which is 18 m high on the south-west side; Palazzo Paleotti on the south-east and south side, which is approximately 13–15 m high. The sky view factor of Piazza Verdi square is small, which means that there is a limited surface directly irradiated by the sun.

On the other hand, Piazza Scaravilli is surrounded by a portico of 4 m high and a building of 8 m high, while another building of 13 m high does not directly face the square but is nearby. Thus, in this case, a wide surface is directly irradiated by solar radiation.

Another effect depends on the boundary area of the simulation model: Piazza Verdi boundaries include a garden, called Giardino del Guasto, which is a green area, with vegetation and high trees, located at the north of the theatre. Conversely, in the case of Piazza Scaravilli, there are no green surrounding areas, which explains the higher air temperature values and the worst PET values, as the ACCIONA sensors also confirm.

Finally, street and square pavements have a considerable influence on albedo (or reflectance) due to mineral materials. Piazza Verdi and the adjacent streets have a homogenous pavement with an albedo of nearly 0.40, while Piazza Scaravilli has several types of street materials: asphalt (albedo 0.20) and two other types of mineral pavements with albedos of 0.4 and 0.80 under the portico. In these areas, as it often happens in historic city centres, the ceiling is mainly characterised by the presence of mineral pavements, while diverse kinds of pavement (e.g., brick, ceramic, etc.) or green areas are substantially absent.

To summarise, from the analysis of the data, it can be said that the microclimate can have a role in people's presence. From the comparison between the number of people on days with and without a public event, it is clear that the event is the factor that determines the crowding (Table 8 and Figure 12). However, outdoor microclimatic conditions at other times of the day, for example, at 1 pm, as shown in Table 11, prove that there is still a relationship between PET and the number of people. Although the increase in people is not high (between 50 and 200) for both pairs of days (26–27 June and 16–17 August) and for both areas (Piazza Scaravilli and Verdi), Envi-met simulations show a direct relationship with PET value: better thermal comfort conditions (low PET values) correspond to a higher people flow in the squares.

**Table 11.** Comparison of the number of people present in both squares in relation to the simulation results of PET and Visitors Count at 13:00 h, both areas.

| Area | Date | PET (°C) | Persons (Num.) |
|---|---|---|---|
| Piazza Verdi | 26th of June | 26–27 | 2500 |
| | 27th of June | 29–30 | 2400 |
| | 16th of August | 20–21 | 700 |
| | 17th of August | 22–23 | 600 |
| Piazza Scaravilli | 26th of June | 46–48 | 1400 |
| | 27th of June | >48 | 1200 |
| | 16th of August | 38–40 | 250 |
| | 17th of August | 40–42 | 200 |

All considerations are empirical because it is not possible to express the direct correlation between microclimate conditions and people's flow. The reasons why people move are

different and multiple. In this case, it was possible to correlate microclimatic conditions with a precise cause of people's movement, namely the event.

## 7. Conclusions and Future Developments

The paper aims to relate the results of microclimatic simulations to the presence of people in public spaces. In particular, the study focuses on Piazza Verdi and Scaravilli, located in the historic centre of the city of Bologna, with the aim of investigating the correlation between microclimatic factors (air temperature, wind speed, relative humidity) and perceived thermal comfort (PET), evaluated thanks to the use of Envi-met software, and people flows, detected through sensors.

The aim of the research is to assess whether the presence of people is linked exclusively to the occurrence of events, or if specific microclimatic conditions can significantly affect the choice of people to stay in the area in question. Finally, our research responds to the question of whether a direct correlation exists between outdoor microclimate in public space and people's presence and if a public event plays a role in altering it. The results show that such a correlation exists, and our research methodology allows us to study it, making it possible to replicate our method in future research.

The research has chosen to simulate the pairs of days of 26–27 June and 16–17 August, since both include a day on which a public event has occurred (26th June, during the performance by Neri Marcorè, and 17 August, during a public performance of Tosca) and a day without events (respectively 27 June and 16 August). This has made it possible to analyse whether the increase in visitors depended only on the event or if the microclimatic conditions also had an impact on the attendance at outdoor spaces.

The research has some limitations: people's presence does not depend solely on outdoor microclimate and on events, but several factors can play a role in people's presence in the study areas, e.g., weather conditions (e.g., rain, etc.), holidays and weekdays (the weekend is likely to be more crowded). For this reason, our simulation accounts for days with and without public events. We know the complexity of motivations, so our other ongoing research aims to improve this physics analysis by also using multiple indicators that combine qualitative and quantitative indicators. This paper constitutes the quantitative starting point of a more qualitative analysis that will be conducted in the future.

In conclusion, the research provides a method of analysis that, by merging data about microclimate and crowd flows, allows one to gain a deeper knowledge of outdoor thermal comfort in open spaces and to act in order to improve the outdoor thermal comfort. However, it is important to highlight how outdoor thermal comfort in urban open spaces is determined by urban elements such as squares, streets, neighbourhoods, districts, etc. Meteorological conditions influence the microclimate, but the architectural and urban spatial configuration, including the surface materials, play a crucial role in determining the outdoor thermal comfort.

### Future Studies

Finally, even if an analysis of the role of the crowd on microclimate would be topical, the Envi-met software does not allow for the evaluation these types of data. Therefore, it was not possible to evaluate the role of the crowd in the microclimate. However, this issue can be developed in future research and studies. Moreover, our aim, in this research, is to study the correlation between people's presence in public spaces during public events and their use on days without events. We consider PET as the best indicator to evaluate it. Thus, we considered the influence of MRT in PET, but the impact of MRT and solar levels on people's presence should be improved in future research.

Also encouraged by the research and results described in this paper, the municipality and the University of Bologna have carried out temporary regeneration actions experiments that involved public spaces in the university area as part of the activities implemented in the ROCK project framework. These experiments were not monitored with the methodology illustrated in this article, but citizen satisfaction pushed the Bologna municipal administra-

tion to undertake a transformation path of these spaces into a resilient and inclusive area, moving towards a series of incremental interventions aimed at improving microclimatic conditions, outdoor comfort and their general livability [73]. It is possible that, despite the fact that the ROCK project ended in December 2020, future monitoring opportunities will be activated to evaluate the impacts of such interventions.

**Author Contributions:** Conceptualization, K.F., A.B. and D.L.; methodology, K.F. and D.L.; software, K.F.; validation, K.F. resources, A.B. and D.L.; writing—original draft preparation, K.F., A.B. and D.L.; writing—review and editing, S.B., K.F. and R.R.; supervision, A.B., D.L. and K.F.; project administration, D.L. and R.R.; funding acquisition, D.L., A.B., S.B. and R.R. All authors have read and agreed to the published version of the manuscript.

**Funding:** ROCK (Regeneration and Optimization of Cultural heritage in Creative and Knowledge cities) project is cofinanced by the European Union within the H2020 Framework Programme [ROCK G.A. No. 730280], in the axis "Climate Action, Environment, Resource Efficiency and Raw Materials", "Greening the Economy", in response to the call "Cultural Heritage as a driver for sustainable growth" (Call ID: H2020-SC5-2016-2017).

**Acknowledgments:** A special thanks goes to the ROCK project partners and especially to DFRC, ACCIONA and CORVALLIS.

**Conflicts of Interest:** The authors declare no conflict of interest.

## Nomenclature

| | |
|---|---|
| APIs | Application Programming Interface |
| ARPAER | Agenzia Prevenzione Ambiente Energia Emilia Romagna |
| CH | Cultural Heritage |
| DFRC | Data Fusion Research Centre, http://www.dfrc.com.sg/ accessed 2 May 2023 |
| ICT | Information and Communication Technologies |
| OMM | Outdoor Microclimate Map |
| PET | Physiological Equivalent Temperature |
| PMV | Predicted Mean Vote |
| ROCK | Regeneration and Optimization of Cultural Heritage project in Creative and Knowledge cities |
| SVF | Sky View Factor |
| UHI | Urban Heat Island |
| UTCI | Universal Thermal Climate Index |

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
