# Peer review of "The Relation between Outdoor Microclimate and People Flow in Historic City Context the Case Study of Bologna within the ROCK Project"

_sustainability, doi:10.3390/su15097527_

Round 1

Reviewer 1 Report

The paper is well written and presented. The scope of study is well-established to contribute on thermal comfort studies globally. Therefore, the paper is required additional work in order to improve the credibility of this rigorous research work. I outlined my recommendations as follow:

  1. Improve Abstract to give less context and more on the knowledge gap, research questions, methods and key findings
  2. In Introduction section, please outline the main aim, objectives and research questions clearly and articulate the research questions to implement the neutral adaptive thermal comfort thresholds as an output of the simulation analysis conducted in the study.
  3. In Introduction section, Novelty of the study should be explained.
  4. The authors have been discussed the previous scholars’ work in the Introduction but this is not sufficient to support the research outcomes presented in the Results section. I recommend to the authors to open-up a new section and consider these literature types as follows; systematic literature review or comprehensive literature review to study worldwide literature on thermal comfort studies 
  5. I recommend to the authors to the use this open-source software to conduct the systematic literature review on thermal comfort studies effectively. Here is the link of the open-source software tool - https://www.vosviewer.com - The authors generated the selected keywords and import the data into this software which allows the researcher to generate the visual maps. I believe that this tool could increase the scientific credibility of their research work
  6. Materials and Methods section should be re-conceptualised, the authors should provide more detail on technical specifications of research instruments used for the field study.
  7. In Sub-section 4.1 (Outdoor physics variables) section should refer more similar pilot studies to demonstrate the significance of authors’ their own work. I recommend to the authors to read this article - Ozarisoy, B., & Altan, H. (2021). Regression forecasting of ‘neutral’ adaptive thermal comfort: A field study investigation in the south-eastern Mediterranean climate of Cyprus. Building and Environment, 202. https://doi.org/10.1016/j.buildenv.2021.108013 - To increase the credibility of the authors’ their own work, I recommend the authors to cite this article while they are referring their own methodological framework in thermal comfort studies 
  8. In Sub-section 4.1, the statistically representativeness of the sample size should be discussed. Is this sample size sufficient to make a generalisation of the study findings in thermal comfort studies? 
  9. In Sub-section 4.1, the method should give an honest appraisal of how the sample size were chosen and reference the work of others who have developed statistically representative archetypes 
  10. 10. In sub-section 4.1, With regards to the identify the statistically representativeness of the sample size, please consider the statistical power of the survey. Use this open-source tool to identify the appropriate type of statistical method. Here is the link - https://www.psychologie.hhu.de/arbeitsgruppen/allgemeine-psychologie-und-arbeitspsychologie/gpower - This is the power analysis tool which helps you to identify the appropriate type of statistical method. Please use this tool and revise the statistical results presented in the Results section. 
  11. In section 5.1 (Env-Met results calibration), The authors also advised to link their results to the previous work shared in the ASHRAE global comfort database II which is available online throughout this link - http://www.comfortdatabase.com/ .
  12. The authors research work is noteworthy contribution the the ASHRAE Global Thermal Comfort Database II - Please donate your dataset to this link - https://datadryad.org/stash/dataset/doi:10.6078/D1F671
  13.  In section 5.1, Comparing with the result from the ASHRAE global thermal comfort database II with the same region might be interesting 
  14. In section 5.2, the air velocities are completely neglected. Whereas they are significant to restore the thermal comfort. Please provide more experiments in the Results section to demonstrate 
  15. In this work, the indoor movement is exempted. In fact, the indoor air movement will have a high impact for reaching neutral thermal sensation. It can make the occupants feel comfortable in relatively high indoor temperature
  16. 16. The reviewer afraid that these results cannot be generalised for the whole population sample due to the respondent limitation which can generate results. Please provide more evidence on this matter in order to increase the scientific soundness of this paper. I recommend to read the this article and make the comparison analysis within your findings. (1)   Altan, H., & Ozarisoy, B. (2022). Dynamic evaluation method for assessing households’ thermal sensation using parametric statistical analysis: A longitudinal field study in the South-eastern Mediterranean climate. CLIMA 2022: the 14th REHVA HVAC World Congress. Rotterdam, NL 22-25 May 2022, CLIMA 2022. https://doi.org/10.34641/clima.2022.422 - To increase the credibility of this article, I recommend you to cite this article properly in your discussions section.
  17. In discussion, There are three thermal adaptation types. They are physiological, which is related to the body reaction due to the temperature change, psychological which is derived from the state of mind of previous experiences and behaviour related adaptation (Brager and de Dear, 1998). Comfort can be reached if there are sufficient opportunities for people to adapt. The comfortable temperature is changeable rather than fixed. (Fergus Nicol and Roaf, 2015). Please consider these theoretical information while you are interpreting the statistical analysis. 
  18. Relate your conclusions to your research questions

Author Response

Dear Reviewer,
we really appreciated your comments and advices; therefore, according to these, the whole manuscript has 
been revised and improved. 
We hope that our last version obtain a positive evaluation.
Kind regards 
Kristian Fabbri

Reviewer 2 Report

This study presents aspects of relation between outdoor microclimate and people flow.

The paper requires major revision to meet the requirements of scientific journals having high IF.

1.     Please explain clearly what is novelty in this paper.

2. The innovation and scientific contribution of this study need to be highlighted at the end of the introduction and the conclusions.

2.     Please explain criteria of the analysis and measurement methods, criteria of choosing measuring parameters.

3.     The conclusion section should be adapted such that it is totally supported by the obtained results. It is necessary to add the conclusions and add possibility of application.

4.     I recommend this article to publication in Sustainability after major revision.

Author Response

Dear Reviewer,
we really appreciated your comments and advices; therefore, according to these, the whole manuscript has been revised and improved.  We hope that our last version obtain a positive evaluation.
Kind regards 
Kristian fabbri

Reviewer 3 Report

This is a refreshing paper to read. The questions are interesting and convincingly investigated by an appropriate methodology. I am not an expert in ENVImet and I can therefore not comment on some of the data results in great detail. However, it seems to be clearly presented and the results seem to make sense. The conclusions are also clearly presented based on the results from the data and simulations. It could be good to include potential future research in the discussion section. Where could further studies go related to this research?

One key point is that the English should go through another thorough round of editing. This can be done with something like Grammarly but probably also with someone to proof read it. There were a few things I noticed:

- Avoid informal, conversational language: for example, in journal articles, it should be "did not" rather than "didn't"

- Check precision or meaning of terms in English. For example, what is meant by: “physics of an open space”? Physical dimension of the space? Physics could include a lot of things. It’s a whole scientific discipline.

Author Response

Dear Reviewer,
we really appreciated your comments and advices; therefore, according to these, the whole manuscript has been revised and improved.  We hope that our last version obtain a positive evaluation.
Kind regards, Kristian Fabbri

AUTHOR REPLY
This is a refreshing paper to read. The questions are interesting and convincingly investigated by an appropriate methodology. I am not an expert in ENVImet and I can therefore not comment on some of the data results in great detail. However, it seems to be clearly presented and the results seem to make sense. The conclusions are also clearly presented based on the results from the data and simulations. It could be good to include potential future research in the discussion section. Where could further studies go related to this research?
#REPLY:. We report a future results section in conclusion.

One key point is that the English should go through another thorough round of editing. This can be done with something like Grammarly but probably also with someone to proof read it. There were a few things I noticed:

- Avoid informal, conversational language: for example, in journal articles, it should be "did not" rather than "didn't"
- Check precision or meaning of terms in English. For example, what is meant by: “physics of an open space”? Physical dimension of the space? Physics could include a lot of things. It’s a whole scientific discipline.
#REPLY: Amended

Reviewer 4 Report

Dear authors,

The research on the relationship between public participation and public events due to climatic aspects, needs further clarification; the results presented correspond to general and common sense perceptions.

Author Response

Dear Reviewer,
we really appreciated your comments and advices; therefore, according to these, the whole manuscript has been revised and improved.

#REPLY:. We adjunct a novelty of the study at the end of introduction paragraph in order to respond to you.

We hope that our last version obtain a positive evaluation.
Kind regards 

Round 2

Reviewer 1 Report

The authors have been accepted all changes very thoroughly and the paper is ready for a consideration of publication.